# Selective Hepatic *Cbs* Knockout Aggravates Liver Damage, Endothelial Dysfunction and ROS Stress in Mice Fed a Western Diet

**DOI:** 10.3390/ijms24087019

**Published:** 2023-04-10

**Authors:** Sebastiaan Lambooy, Andries Heida, Christian Joschko, Dalibor Nakladal, Azuwerus van Buiten, Niels Kloosterhuis, Nicolette Huijkman, Albert Gerding, Bart van de Sluis, Robert Henning, Leo Deelman

**Affiliations:** 1Department of Clinical Pharmacy and Pharmacology, University Medical Center Groningen, University of Groningen, 9713GZ Groningen, The Netherlands; 2Department of Pediatrics, University Medical Center Groningen, University of Groningen, 9713GZ Groningen, The Netherlands; 3Department of Laboratory Medicine, University Medical Center Groningen, University of Groningen, 9713GZ Groningen, The Netherlands

**Keywords:** cystathionine-β-synthase, hyperhomocysteinemia, hydrogen sulfide, knockout mice, high-fat diet, reactive oxygen species, liver damage, NAFLD

## Abstract

Cystathionine-β-synthase (CBS) is highly expressed in the liver, and deficiencies in *Cbs* lead to hyperhomocysteinemia (HHCy) and disturbed production of antioxidants such as hydrogen sulfide. We therefore hypothesized that liver-specific *Cbs* deficient (LiCKO) mice would be particularly susceptible to the development of non-alcoholic fatty liver disease (NAFLD). NAFLD was induced by a high-fat high-cholesterol (HFC) diet; LiCKO and controls were split into eight groups based on genotype (con, LiCKO), diet (normal diet, HFC), and diet duration (12 weeks, 20 weeks). LiCKO mice displayed intermediate to severe HHCy. Plasma H_2_O_2_ was increased by HFC, and further aggravated in LiCKO. LiCKO mice fed an HFC diet had heavier livers, increased lipid peroxidation, elevated ALAT, aggravated hepatic steatosis, and inflammation. LiCKO mice showed decreased L-carnitine in the liver, but this did not result in impaired fatty acid oxidation. Moreover, HFC-fed LiCKO mice demonstrated vascular and renal endothelial dysfunction. Liver and endothelial damage correlated significantly with systemic ROS status. In conclusion, this study demonstrates an important role for CBS in the liver in the development of NAFLD, which is most probably mediated through impaired defense against oxidative stress.

## 1. Introduction

Hyperhomocysteinemia or HHCy is an established risk factor for the development of cardiovascular disease, chronic kidney disease, and Alzheimer’s disease [1,2,3,4,5]. In addition, it has been demonstrated in a number of meta-analyses that HHCy is associated with non-alcoholic fatty liver disease (NAFLD), steatohepatitis, as well as all-cause mortality [6,7]. Severe cases of HHCy are frequently associated with mutations in the gene encoding for cystathionine-β-synthase (CBS), with an incidence of approximately 1 in 100,000 in western countries [8], whereas mild cases of HHCy may be caused by processes which affect vitamin B uptake, resulting in vitamin B12 deficiency [9].

CBS is a lyase-class enzyme that is predominantly expressed in the liver. It is the rate-limiting enzyme in the transsulfuration pathway where it is responsible for the metabolic conversion of homocysteine to the amino acid cystathionine [10]. Moreover, CBS can produce the gasotransmitter H_2_S in a reaction between cysteine and homocysteine. H_2_S is involved in a plethora of cellular processes including regulation of oxidative stress, glucose, lipid metabolism, and mitochondrial and vascular function [11,12,13,14,15]. In addition, pathways downstream of CBS include glutathione and taurine biosynthesis, which are both antioxidants [6,7].

To investigate the role of CBS in HHCy, several mouse models of CBS deficiency have been developed. The classical homozygous *Cbs* knockout mice were developed by Maeda and colleagues, and demonstrated high perinatal mortality associated with liver failure from severe fibrosis and neutrophil infiltration [16]. Surviving *Cbs* knockout mice were characterized by HHCy, steatotic hepatomegaly, and early liver fibrosis with elevated levels of alanine aminotransferase (ALAT), tumor necrosis factor alpha (TNF-α), and interleukin (IL-6) in plasma. To overcome the problems of perinatal mortality in the Maeda model, the *Tg-hCBS I278T* mouse model was developed, in which a mutated human *CBS* gene under the control of a zinc-sensitive promotor was introduced, enabling rescue of embryonic LiCKO mice by the addition of zinc to the drinking water [17]. The *Tg-hCBS I278T* mouse showed HHCy and less liver damage than the Maeda model. In addition, the HO (human only) *Cbs* knockout mouse was developed, in which a copy of the complete human CBS gene was included in the germ line followed by crossing with the Maeda knockout mouse [18]. HO *Cbs* knockout mice lacked neonatal lethality, demonstrated variable plasma homocysteine concentrations, and showed mildly increased ALAT levels, but did not suffer from hepatic steatosis. Taken together, these studies demonstrate a protective role for CBS in the liver, particularly during the perinatal period.

To circumvent potential problems with perinatal mortality, our group recently developed a conditional *Cbs* knockout mouse model based on the Cre-ERT2/LoxP recombination system that is inducible by tamoxifen, which allowed us to inactivate *Cbs* at an adult age. Treatment with tamoxifen at an age of 10 weeks resulted in inactivation of *Cbs* in all investigated organs and tissue (liver, kidney, and skin) but not in the brain [19]. In a study of ageing, these conditional *Cbs* knockout mice showed a mild phenotype with impaired growth, facial alopecia, endothelial dysfunction but not increased mortality, nor signs of liver or kidney damage, despite severe HHCy. Moreover, animals demonstrated increased serum hydrogen peroxide levels and increased hepatic expression of MnSOD, indicating a higher oxidative load. As HHCy was not directly linked to liver development in this study, HHCy may be indicative of problems further downstream in the transsulfuration pathway under conditions of CBS deficiency. This may result in impaired cysteine conversion, reduced H_2_S production, and decreased glutathione synthesis. As these pathways are highly involved in defence against reactive oxygen species in the liver, we hypothesize that liver-specific *Cbs* knockout mice would be particularly susceptible to the development of liver damage in models with high oxidative stress.

To further investigate the role of CBS in the liver, here we report on the development of a liver-specific *Cbs* knockout mouse (LiCKO) by breeding the previously developed *Cbs*^flox/flox^ mice with hemizygous Alb-cre mice. Moreover, we explored the effects of oxidative stress on liver damage and associated systemic damage by feeding mice a Western diet (high fat, high cholesterol).

## 2. Results

### 2.1. Model and Confirmation of Selective Deletion of CBS in Liver

Breeding of homozygous *Cbs*^flox/flox^ mice with hemizygous Alb-cre mice resulted in normal litter-sizes (*n* = 7.6 ± 0.4) with expected mendelian distribution of genotypes, indicating that problems with perinatal mortality seen in previous models [16] were not present. Western blotting for CBS in liver and kidney lysates in a subset of animals confirmed the nearly complete deletion of CBS in the liver but not in kidneys (Figure 1a,b, respectively). Moreover, H_2_S production was almost completely blunted in livers of LiCKO mice compared with controls (Figure 1c).

Amino acid analysis (Appendix A) of liver lysates further confirmed the inhibition of CBS, as cystathionine levels were below the lower limit of detection in LiCKO in contrast to controls (47 ± 19 ng/g). Moreover, CBS inhibition also had effects upstream of the transsulfuration pathway, as methionine levels were significantly increased in LiCKO mice compared with controls (456 ± 76 and 376 ± 90 ng/g, respectively). In addition, LiCKO mice demonstrated increased threonine (1270 ± 172 vs. 1084 ± 239 ng/g), proline (597 ± 96 vs. 491 ± 137 ng/g) and glycine (4068 ± 500 vs. 3711 ± 295 ng/g) when compared with controls.

Relative GSH levels were slightly lower in LiCKO mice on ND than controls on ND (86 ± 5 and 100 ± 1%, respectively). However, GSH levels in controls on HFC and LiCKO on HFC showed no further decrease (82 ± 7 and 81 ± 3%, respectively).

### 2.2. Main Animal Characteristics

To assess the effects of HFC on LiCKO mice, control and LiCKO mice were placed on either ND or HFC for a period of 12 or 20 weeks. The main characteristics of all mice are shown in Figure 2 and Table 1. LiCKO mice on an ND were heavier than control mice at 12 weeks of diet (Figure 2a, 31.8 ± 0.8 g and 29.0 ± 0.4 for LiCKO and con, respectively) and 20 weeks of diet (Figure 2b, 32.7 ± 0.9 and 29.3 ± 1.0 g for LiCKO and con, respectively). Mice on HFC were heavier than mice on ND, but no differences in bodyweight could be observed between genotypes. Nuclear magnetic resonance (Appendix A) did not show major differences between genotypes, except for a slightly higher body fat percentage in LiCKO mice on ND over controls on ND at 20 weeks of diet (23.8 ± 5.1 and 16.4 ± 2.6%, respectively). HFC resulted in a dramatic increase in body fat percentage that was similar in controls and LiCKO.

After termination, organs and blood were collected. At 12 weeks of HFC diet, a trend towards increased liver weight could be observed in LiCKO over control mice on HFC (Table 1 and Figure 2c). This difference became statistically significant at 20 weeks of HFC diet in LiCKO mice (2167 ± 205 mg) compared with controls on HFC (1401 ± 89 mg) (Figure 2c, *p* < 0.05). Weights of other organs did not differ between treatments and genotypes, although we observed one mouse in the LiCKO group with splenomegaly (spleen weight 261 mg) at 20 weeks of HFC diet.

Hematological analysis showed significant differences only at 20 weeks of diet (Table 1, Figure 2), with LiCKO mice on HFC showing increased white blood cell count (Figure 2e,f; WBC, 6.0 ± 0.6 vs. 2.9 ± 0.5 × 10^9^/L for con HFC), red blood cell count (RBC, 11.7 ± 0.4 vs. 9.8 ± 0.1 × 10^12^/L for con HFC), hemoglobin (HGB, 10.2 ± 0.4 vs. 8.8 ± 0.1 mM for con HFC), and hematocrit (Figure 2g,h; HCT; 0.59 ± 0.02 vs. 0.49 ± 0.00 L/L for con HFC). In contrast, platelet count (PLT) was decreased in LiCKO mice at 20 weeks of HFC (Figure 2i,j; 856 ± 70 vs. 1161 ± 108 × 10^9^/L for con HFC).

### 2.3. LiCKO Mice Have Mild to Severe HHCy

HHCy was present in all LiCKO mice (Figure 3a,b), with some differences depending on diet. At 12 weeks of diet, plasma homocysteine in LiCKO mice on ND was 76.2 ± 3.2 µM whereas in LiCKO mice on HFC this was significantly higher at 117.7 ± 6.8 µM (*p* < 0.05). At 20 weeks of diet, plasma homocysteine levels were similarly elevated for LiCKO on both ND and HFC (75.6 ± 6.7 µM and 88.8 ± 5.2 µM, respectively). Control mice demonstrated low homocysteine levels (<10 µM) on both ND and HFC at 12 weeks and 20 weeks of diet.

### 2.4. LiCKO Mice on HFC Exhibit Elevated Circulating ROS and Hepatic ROS Damage

In order to determine the systemic ROS status of the mice, we measured plasma levels of hydrogen peroxide using an amplex red assay (Figure 3c,d). Two-way ANOVA analysis demonstrated significant effects of both diet (F (1, 28) = 41.69, *p* < 0.0001) and genotype (F (1, 28) = 19.042, *p* = 0.0002) on plasma H_2_O_2_ levels at 12 weeks of diet. Similar results were obtained for plasma H_2_O_2_ levels at 20 weeks of diet: F (1, 26) = 46.76, *p* < 0.0001 and genotype: F (1, 26) = 5.497, *p* = 0.027.

To assess ROS damage in the liver, a slot-blot analysis was performed on liver lysates using an anti-malondealdehyde (MDA) antibody (Figure 3e,f). Although the data showed considerable variation at 12 weeks of diet, a significant increase in liver MDA staining was observed in LiCKO mice on HFC at 20 weeks compared with control mice on HFC (*p* < 0.05).

### 2.5. LiCKO Mice on HFC Exhibit Elevated Circulating ALAT on HFC

To investigate the effect of hepatic deletion of CBS on liver damage markers, we measured alanine aminotransferase (ALAT) activity in plasma (Figure 3g,h). ALAT activity was significantly increased in LiCKO mice at both 12 and 20 weeks of HFC (175 ± 8 and 240 ± 6% of control, respectively). ALAT activity in LiCKO mice on ND and controls on HFC did not differ from controls on ND.

### 2.6. Liver Histology Shows Aggravated Pathology in LiCKO Mice on HFC

Hematoxylin and eosin staining was performed on liver sections obtained after termination (Figure 4). The main histological pattern observed in all groups on HFC (Figure 4c,d,g,h) was characterized by liver steatosis from centrolobular to azonal, both micro- and macrovesicular, lobular inflammation, and minimal to moderate oval cell hyperplasia in the portal/periportal region. Steatotic lesions were more severe in LiCKO mice at 12 and 20 weeks of HFC compared with the corresponding control groups on HFC, and were especially severe in LiCKO at 20 weeks of HFC. To quantify the severity of fatty liver disease, we calculated NAFLD activity scores (NAS, Figure 4i,j), a composite of scores for steatosis, ballooning, and lobular inflammation (Appendix A). At 12 weeks of diet, NAS was significantly increased by HFC diet in controls as well as in LiCKO (2.3 ± 0.5 and 3.8 ± 2.3, respectively) compared with respective groups on ND (0.4 ± 0.5 and 1.4 ± 0.5, respectively), without significant differences between genotypes. At 20 weeks of diet, NAS was significantly further increased in LiCKO mice on HFC diet (5.1 ± 1.4) compared with controls on HFC (3.3 ± 1.4).

### 2.7. Liver Triglyceride and Cholesterol Levels Confirm Aggrevated Steatosis in LiCKO Fed a HFC

To confirm the liver steatosis observed in histological sections, we determined triglyceride, free cholesterol, total cholesterol, and cholesteryl ester content in liver samples (Figure 5). At 12 weeks of diet (Figure 5a), triglyceride levels in liver tissue were similarly increased in controls fed a HFC (26.3 ± 22 µmol/g) and in LiCKO mice fed an HFC (49 ± 26 µmol/g) compared with controls on ND (2.7 ± 0.8 µmol/g) and LiCKO on ND (5.9 ± 2.7 µmol/g), respectively. At 20 weeks of diet (Figure 5b), triglyceride levels in liver tissue were further increased in LiCKO mice fed an HFC diet over controls fed an HFC diet (64 ± 23 and 39 ± 10 µmol/g, respectively). Free cholesterol did not differ between groups (Figure 5c,d). Total cholesterol (Figure 5e,f) was found to be further increased in LiCKO mice fed an HFC compared with controls fed an HFC at both time points (week 12: con HFC 7.5 ± 4.3, LiCKO HFC 18.5 ± 10.3; week 20: con HFC 10.5 ± 3.7, LiCKO HFC 19.3 ± 8.7 µmol/g). Similarly, cholesteryl ester (Figure 5g,h) was found to be further increased in LiCKO mice fed an HFC compared with controls fed an HFC at both time points (week 12: con HFC 5.6 ± 3.6, LiCKO HFC 16.0 ± 9.5; week 20: con HFC 8.5 ± 3.3, LiCKO HFC 16.5 ± 7.8 µmol/g).

### 2.8. Dyslipidemia Is Not Aggrevated in LiCKO Mice Fed an HFC Diet

To assess whether dyslipidemia was affected in LiCKO mice fed an HFC diet, plasma levels of triglycerides and cholesterol were determined (Appendix A). Triglyceride levels did not differ between groups (Appendix A). Plasma cholesterol levels were increased by HFC diet at both time points (Appendix A), without significant differences between controls and LiCKO. (Week 12: con HFC 5.2 ± 1.8, LiCKO HFC 4.7 ± 0.7; week 20: con HFC 4.9 ± 0.8, LiCKO HFC 5.6 ± 1.0 mmol/L).

### 2.9. LiCKO Mice Have Decreased L-Carnitine Levels in the Liver

Impaired fatty acid oxidation resulting from low L-carnitine levels has been implicated in the development of liver steatosis. We therefore determined L-carnitine levels in liver lysates (Figure 6). At both 12 and 20 weeks of diet, we found primarily carnitine C0 and C2 with substantially lower expression than the other carnitines (Appendix A). At 12 weeks of diet (Figure 6a), total, C0, and C2 acylcarnitine levels were lower in LiCKO mice fed HFC (37 ± 15, 27 ± 12, 4.9 ± 1.7 nmol/g, respectively) compared with controls on HFC (125 ± 35, 84 ± 24, 23 ± 9 nmol/g, respectively). Moreover, C2 was also lower in LiCKO mice on an ND diet (19.3 ± 9.2) versus controls on ND (38 ± 18, 19 ± 9 nmol/g, respectively). At 20 weeks of diet (Figure 6b), total and C0 acylcarnitines were lower in LiCKO mice compared with controls regardless of diet. C2 acylcarnitine at 20 weeks was lower in LiCKO on ND (7.6 ± 4.6 nmol/g) versus controls on ND (32.6 ± 23 nmol/g) with no differences between groups on an HFC diet.

### 2.10. LiCKO Mice Do Not Show Signs of Impaired Fatty Acid Metabolism

To investigate whether decreased acylcarnitine levels in the liver were associated with decreased fatty acid metabolism, we investigated mitochondrial respiration (Oroboros) in isolated mitochondria from fresh liver tissue, using palmitoyl-carnitine as substrate, in a subset of animals. We did not find differences in basal respiration, state 3, or full uncoupling between controls and LiCKO mice at 20 weeks of HFC diet (Appendix A). In addition, mice were studied by multiplexed respirometry in Phenomaster cages (TSE) at 12 weeks of diet (Appendix A). Mice on ND showed clear circadian rhythms in respiratory exchange ratio (RER 1.0 active, 0.8 resting) and metabolism (0.4 kcal/h active, 0.25 kcal/h resting) with no differences between controls and LiCKO. Mice on HFC had lower RER (0.85–0.9) in the active phase compared to mice on ND, indicating mixed metabolism of both fat and carbohydrates. RER in the resting phase was 0.8 for controls and LiCKO mice on HFC diets. No differences between controls and LiCKO were observed in mice on HFC diets. Taken together, these data indicate no clear differences in fatty acid metabolism between LiCKO mice and controls.

### 2.11. HFC Causes Vascular and Renal Endothelial Dysfunction in LiCKO Mice

To investigate potential systemic effects of the liver-specific deletion of *Cbs*, we investigated vascular and renal endothelial function. Specifically, we assessed endothelium-dependent relaxation (EDR) in aortic rings and measured the albumin–creatinine ratio (ACR) in urine. For this, the thoracic descending aorta was preserved in cold saline solution, cut into rings, and mounted in a Mulvany myograph. EDR was measured in rings preconstricted with phenylephrine (PE) with subsequent stimulation by acetylcholine (Figure 7). HFC diet did not affect total EDR in control mice at 12 or 20 weeks of diet (Figure 7a,c, respectively). Interestingly, HFC diet reduced EDR in LiCKO mice at both 12 and 20 weeks compared to ND (Figure 7b,d). Quantification of the area under the curve (AUC) demonstrated a significant decrease (*p* < 0.05) in total EDR for HFC in LiCKO mice at week 12 (109 ± 4 vs. 176 ± 6 AUC for HFC and ND, respectively) and week 20 (127 ± 3.5 vs. 183 ± 4 AUC, for HFC and ND, respectively). ACR at 12 weeks of diet did not reveal differences between groups. Moreover, ACR (Figure 7e) was significantly increased at 20 weeks of HFC diet in LiCKO mice when compared with controls on HFC (103 ± 33 and 55 ± 37 µg/mg, respectively). Taken together, these data indicate that both the vascular and renal endothelium are affected by HFC in LiCKO.

### 2.12. Systemic Oxidative State Is Correlated with Liver, Vascular, and Renal Damage

To assess whether oxidative stress correlated with the observed liver damage and endothelial damage, we constructed regression plots of ALAT activity (Figure 8a), EDR (Figure 8b), and ACR (Figure 8c) versus serum hydrogen peroxide concentrations in individual animals. ALAT and ACR correlated positively with hydrogen peroxide, indicating more liver and kidney damage with increasing hydrogen peroxide concentrations. EDR correlated negatively with serum hydrogen peroxide, indicating decreasing EDR with increasing hydrogen peroxide concentrations.

## 3. Discussion

The current study demonstrates that selective genetic deletion of CBS in the liver leads to aggravated liver damage in mice fed a HFC diet. Although LiCKO mice demonstrated decreased L-carnitine in the liver, we did not observe changes in fatty acid oxidation. Rather, the aggravated liver damage in LiCKO mice fed HFC may be related to the loss of hydrogen sulfide production in the liver with subsequent impaired defence against reactive oxygen species. These effects were not limited to the liver, but extended to endothelial dysfunction in the vasculature and kidney.

To our knowledge, this is the first study reporting organ-specific genetic deletion of CBS in mice. In our study, HHCy in LiCKO mice on a normal diet did not lead to liver damage. Interestingly, the addition of an HFC diet caused substantial aggravated liver damage in LiCKO as measured by increased liver weight, elevated plasma ALAT activity, steatosis, and lobular inflammation. In contrast, livers of control mice on HFC only showed steatosis, whereas control and LiCKO mice on a normal diet showed no liver pathology. Taken together, these data demonstrate that LiCKO mice do not show a clear phenotype unless they are challenged by HFC, which causes liver damage. Moreover, liver damage was achieved at relatively moderate HHCy (average homocysteine 90 ± 4 µM, range 48–144 µM), challenging the threshold theory according to which homocysteine levels higher than 200 µM are required to cause end-organ damage [17].

Several hematological parameters were affected in LiCKO mice at 20 weeks of HFC, including decreased platelets and increased white blood cell counts. Decreased platelets or thrombocytopenia may be related to portal hypertension with associated splenomegaly [20]. Indeed, we observed one mouse with splenomegaly with moderate thrombocytopenia (435 × 10^9^/L) in the LiCKO group at 20 weeks of HFC diet. Additional mechanisms for observed thrombocytopenia may involve reduced hepatic production of thrombopoietin in NAFLD [21,22]. Moreover, the relative levels of platelet numbers and white blood cells counts would result in a lower platelet to white blood cell ratio (PWR). Low PWR has been studied as a hematologic marker of systemic inflammatory response and is considered a poor indicator of prognosis for acute-on-chronic liver failure [23]. Additional hematological parameters include increased hemoglobin and red blood cell count, which may both relate to the observed increased hematocrit in LiCKO mice at 20 weeks of HFC. Increased hematocrit levels were also found in patients with NAFLD, and were independently associated with fibrosis in NAFLD patients [24]. The underlying mechanisms for increased hematocrit in our study may include a decrease in plasma volume. However, our data on whole body composition do not indicate a difference in free fluid levels between controls and LiCKO mice on an HFC diet. Plasma volume is a considerable portion of the total fluids, and these data do not suggest major differences in plasma volume between controls and LiCKO on an HFC diet. Alternatively, the increased hematocrit in NAFLD may be related to an impaired reticuloendothelial system in the liver, resulting in impaired clearance of old or defective erythrocytes [25]. Taken together, our data on hematological parameters further support the findings of aggravated liver damage in LiCKO mice on HFC.

To investigate the mechanisms involved in the liver damage in LiCKO mice on HFC, we investigated fatty acid metabolism. Carnitines facilitate the entry of fatty acids into the mitochondria, which is the rate-limiting step in fatty acid oxidation. In the present study, we found decreased carnitines in LiCKO, which may point to either increased transport and usage of carnitines or to reduced synthesis of carnitines. Transport of carnitines into the mitochondria has been demonstrated to be mediated through an inhibitory effect of H_2_S on the carnitine/acylcarnitine carrier [26]. Therefore, loss of H_2_S in LiCKO could potentially increase carnitine transport into the mitochondria. Interestingly, we did not find altered fatty acid oxidation in isolated liver mitochondria using palmitoyl-carnitine as substrate, indicating that carnitine transport and fatty acid metabolism in the mitochondria are not affected in LiCKO. However, these findings in isolated mitochondria do not exclude that total fatty acid oxidation capacity may be decreased in the liver of LiCKO mice. Taken together, these data may point to a role for impaired carnitine biosynthesis in the development of aggravated NAFLD in LiCKO. Indeed, carnitine deficiencies have been reported in patients with NAFLD, with improvement after supplementation with carnitine [27,28]. In our study, impaired synthesis of carnitine may explain why we find a general decrease in all investigated carnitines, including free carnitine, short-, and long-chain acyl-carnitines. The mechanisms resulting in impaired carnitine biosynthesis are not fully elucidated and may arise from liver damage itself, as the liver is critical for carnitine biosynthesis [29]. However, we also found decreased carnitine levels in LiCKO mice on ND in the absence of substantial liver damage. Moreover, the decreased carnitine levels in LiCKO mice on ND suggests a direct link between CBS deficiency and carnitine biosynthesis in the absence of disease. At present, the link between CBS and carnitine biosynthesis is unclear. The biosynthesis of carnitine is complex and involves several methylation reactions. As CBS deficiency results in a state of hypomethylation [30], this could impair the methylation reactions involved in carnitine biosynthesis.

Despite the substantially lowered carnitine levels in the liver of LiCKO mice, we did not find differences in whole animals’ fat metabolism between LiCKO and controls on either diet. However, these measurements were performed in conditions of normal solitary housing where mice are not energetically challenged. Increasing their energy expenditure, for example by lowering of ambient temperature, could help to reveal the metabolic effects of lowered carnitines in LiCKO.

Additional roles for L-carnitine have been proposed and recent data indicate a role for L-carnitine in antioxidant defense. This was demonstrated in an in vitro fructose-induced hepatic steatosis model, where L-carnitine attenuated lipid accumulation through AMPK activation by regulating SOD and Nrf2 activity [31]. To further investigate the role of oxidative stress in our study, we investigated systemic oxidative stress (serum hydrogen peroxide) and liver lipid peroxidation (MDA). Serum hydrogen peroxide concentrations and liver MDA immunoreactivity were both increased in LiCKO mice on HFC. Moreover, we found a significant correlation between serum hydrogen peroxide levels and liver damage. Taken together, our data therefore suggest that genetic deletion of *Cbs* may overwhelm the antioxidative defense systems in the liver in mice challenged by HFC, leading to liver damage.

GSH, an important antioxidant downstream of the transsulfuration pathway, has been previously reported to be downregulated (~35–50%) in the liver of mice with genetic deletion of *Cbs* [13,32]. The present study is in line with these findings, although we found only an approximate 15% downregulation of liver GSH in LiCKO mice on ND. However, decreased liver GSH cannot fully explain the exaggerated liver damage observed in LiCKO mice on HFC, as liver GSH levels were found to be equally downregulated both in controls and in LiCKO on HFC. Therefore, these findings may point to an important role for endogenous hydrogen sulfide production in protecting the liver and endothelium against oxidative stress. Indeed, the slow-release hydrogen sulfide donor GYY4137 has been shown to decrease oxidative stress and lipid accumulation and to restore endothelial dysfunction in high-fat-fed apolipoprotein-E-deficient mice [33]. Moreover, GYY4137 alleviated liver damage in high-fat diet (HFD)-treated low-density lipoprotein receptor-knockout (*Ldlr*^−^/^−^) mice [34].

Interestingly, the effects of HFC diet in LiCKO mice were not exclusive to the liver, as additional effects were demonstrated on the vascular and renal endothelium. In the aorta, endothelium-dependent relaxations (EDR) were significantly decreased in LiCKO mice on HFC. In addition, an HFC diet in LiCKO mice was associated with an increase in ACR, indicating damage to the renal endothelium. Moreover, both EDR and ACR showed significant correlation with serum hydrogen peroxide levels, indicating that the vascular and renal damage to the endothelium was related to a systemic increase in ROS stress.

LiCKO mice showed moderate to severe HHCy (average 90 ± 4 µM, range 48–144 µM), which is significantly lower than the HHCy observed in inducible whole body *Cbs* knockout mice (approximately 300 µM) [19]. As CBS expression was almost completely abolished in LiCKO mice, these data point to a considerable contribution of CBS activity in other organs except for the liver, for clearance of homocysteine from plasma. Indeed, CBS expression in adult mice has also been demonstrated in the brain, lungs, and kidneys [19]. However, the contribution of brain CBS activity for maintaining systemic homocysteine levels is minimal, as inducible whole body *Cbs* knockout mice continued to show severe HHCy despite lacking knockout of *Cbs* in the brain.

Interestingly, LiCKO mice thrived on a normal diet and did not show the perinatal mortality and liver damage reported for the classical *Cbs* knockout developed by Maeda and colleagues [16]. This is surprising, as several studies reported the expression of albumin in mouse embryonic liver as early as 10 days post coitus (Theiler stage 16) [35,36], well before the start of CBS expression in the embryonic liver at 12.5–13.5 days post coitus (Theiler stage 20/21) [37,38]. Although there has been some discussion about the onset of the albumin–cre construct, Weisend and colleagues demonstrated that the albumin–cre construct indeed closely matched the expression of endogenous Alb mRNA or protein [39]. Taken together, these data indicate that genetic deletion of *Cbs* in the liver during embryogenesis is not sufficient to induce liver damage as observed in the Maeda model [16].

Our data point to an important role for CBS and H_2_S in limiting oxidative stress in NAFLD. In future studies, we therefore aim to investigate these mechanisms further by employing H_2_S donors such as GYY4137 in our knockout models. Moreover, the decreased carnitine levels in LiCKO mice, even on a normal diet, warrant further investigation into the underlying mechanism.

In conclusion, the current study demonstrates that HHCy is not directly linked to the development of end-organ damage. Rather, HHCy appears to be a proxy for the health of the antioxidant pathways downstream of CBS and suggests an important role for hydrogen sulfide in the defence against oxidative stress. Overwhelming these antioxidant pathways by a HFC diet in a liver-specific CBS knockout mouse may be sufficient for the development of liver damage, which extends to systemic damage to the vascular and renal endothelium. These findings explain why patients with both metabolic disease and HHCy are particularly at risk of developing liver and cardiovascular disease.

## 4. Materials and Methods

### 4.1. Generation of a Liver-Specific Cbs Knockout Mouse Model

*Cbs*^flox/flox^ mice were generated as described previously [19]. The *Cbs*^flox/flox^ mice were subsequently crossed with C57BL/6 mice harboring the Alb-Cre transgene (Jax 003574). Interbreeding of *Cbs*^flox/flox^ Alb-Cre^+/−^ mice generated both LiCKO (liver specific *Cbs*^−/−^, Alb-Cre^+/−,+/+^) and littermate controls (*Cbs*^flox/flox^*,* Alb-Cre^−/−^).

### 4.2. Animal Care and Experimental Protocol

Experiments were approved by the Institutional Animal Care and Use Committee of the University of Groningen, The Netherlands and by the Central Committee for Animal Testing, with approval numbers 15245-02-06 and 186924-01-01. All methods were performed in accordance with local and national guidelines and regulations. Mice were housed solitarily and maintained in an ambient temperature (21 °C)-controlled environment with 12 h/12 h light/dark cycle and ad libitum access to water and normal diet (ND) (diet 801151, RM1 maintenance, Special Diets Services) until the start of the experiment.

A total of 78 males were deployed for the study, evenly distributed into eight groups based on diet (ND vs. HFC), genotype (con vs. LiCKO), and diet duration (12 and 20 weeks). Mice were maintained on either ND or a high fat/high cholesterol diet (HFC, D14010701, Brogaarden, Research Diets, New Brunswick, NJ, USA) for a duration of 12 or 20 weeks. Prior to termination, mice were placed in metabolic cages for 24 h. Urine was collected and metabolic parameters were recorded, including water intake, urine output, and body weight. Whole body composition analysis was performed by time-domain nuclear magnetic resonance (TD-NMR). For this, mice were briefly restrained without anesthesia and placed in a minispec LF series TD-NMR machine (Bruker, Billerica, MA, USA), where fat tissue, lean tissue, and free fluid (plasma, cerebrospinal fluid, and urine) compositions were measured. Anaesthesia was induced in mice by inhalation of 5% isoflurane in oxygen, and mice were humanely terminated by exsanguination. Samples of blood were collected in EDTA tubes, and organs were harvested and split for storage in liquid nitrogen and 4% buffered formaldehyde for fixation (50-00-0 Klinipath-VWR, Amsterdam, The Netherlands).

### 4.3. Mulvany Myography of the Mouse Aorta

During the termination procedure, thoracic aorta and surrounding adipose tissue were surgically removed and placed in cold saline solution (0.9% NaCl, 4 °C). Aortas were carefully dissected from perivascular adipose tissue and cut into 2 mm long rings using a Leica S4E stereoscopic microscope, Vannas stainless steel curved spring scissors with 3 mm blades and Dumont #5 medical biology forceps. Aortic rings were mounted on pins (ø = 200 µm) in a Mulvany multiwire myograph system model 610 M (Danish Myo Technology, Aarhus, Denmark) and incubated at 37 °C in organ baths filled with 5 mL normal physiological Krebs buffer (pH 7.4). Isometric force was sampled, analogue signal was transformed to digital signal using PowerLab 8/30 (ADInstruments, Oxford, UK), and a myogram was recorded in PowerLab Chart v 5.3. After 20 min of equilibration, aortic rings were stretched to transmural pressure of 13.3 kPa using the proprietary DMT normalization procedure software module. Initially, aortic rings were subjected to two subsequent wake-up protocols: addition of 60 mM KCl (7447-40-7, Merck, Schiphol-Rijk, The Netherlands), stabilization of the response, three washes out, and equilibration. Endothelium-dependent relaxation (EDR) was assessed in aortic rings pre-constricted with phenylephrine (PE, 1 µM), and endothelial relaxation responses were recorded in response to cumulative additions of acetylcholine (100 nM–100 μM, A6625, Sigma-Aldrich, Burlington, MA, USA).

### 4.4. Plasma Homocysteine, ALAT and Serum H_2_O_2_ Measurement

After collection of blood in 1.5 mL EDTA tubes, samples were centrifugated at 2000× *g* for 10 min at 4 °C and the supernatant was collected, snap frozen in liquid nitrogen, and stored at −80 °C. Serum was collected by allowing blood to coagulate in normal Eppendorf vials for 20 min at room temperature with subsequent centrifugation at 2000× *g* for 10 min at 4 °C. Total homocysteine (tHCy) levels were determined on an Architect system (Abbott, Hoofddorp, The Netherlands) using a homocysteine reagent kit (ABBL482R02, Abbott, Hoofddorp, The Netherlands). Plasma samples were diluted 25-fold to achieve the required 250 µL sample volume. Plasma homocysteine levels in control groups fell below the lower limit of detection (LLOQ, 1 µM). Given that concentrations of tHCy levels in diluted samples of control groups were below plasma tHCy levels before dilution, the concentration could not have exceeded 25 μM. To establish the average tHCy levels in control groups, samples were pooled and reassessed at a 4-fold dilution. Plasma ALAT levels were measured using the alanine transaminase colorimetric activity assay kit (700260, Cayman Chemical, Ann Arbor, MI, USA). Serum hydrogen peroxide levels were determined using an Amplex red H_2_O_2_ kit, Life Technologies (A22188), Leusden, The Netherlands.

### 4.5. Cholesterol and Triglyceride Analysis in Plasma and Liver

Cholesterol and triglyceride analyses in plasma and liver were performed as previously described [40].

### 4.6. Acylcarnitines and Amino Acids in Livers

Acylcarnitines concentrations were measured in liver homogenate samples, as previously described [41]. Amino acids were measured in liver homogenate samples, as previously described [42].

### 4.7. Urinary Measurements

Urinary albumin levels were determined with a mouse albumin ELISA kit (ab108792, Abcam, Cambridge, UK). Urinary creatinine was measured with the creatinine (urinary) colorimetric assay kit (500701, Cayman Chemical, Ann Arbor, MI, USA).

### 4.8. Measurement of H_2_S Production

Tissue was scraped on dry ice and homogenized using an overhead stirrer at 750RPM (VOS 40 digital, VWR, Amsterdam, The Netherlands) in cold Milli-Q^®^ ultrapure water to prepare a 10% (*w*/*v*) homogenate. The assay reaction mix was formulated to 10 µL volume on a Hard-Shell^®^ 384-well PCR plate (HSP3841, BioRad, Lunteren, The Netherlands), and contained 0.1 M Hepes buffer (pH 7.4), 0.25 mM pyridoxal-5-phosphate (PLP), 2 mM homocysteine, 10 mM cysteine, and 1 µL of liver lysate. To detect H_2_S gas escaping from the liquid phase, we used MN 710 grade filter papers (150047A, Macherey-Nagel™/Fisher Scientific, Landsmeer, The Netherlands) cut to the dimensions of a 384-well plate, soaked in 1% lead (II) acetate trihydrate (Pb(CH_3_COO)_2_ × 3H_2_O, 467863, Sigma-Aldrich) and dried. The plate was shaken using an optical plate reader (Flex Station 3, Molecular Devices, Winnersh, UK) for 10 s and briefly centrifugated. The top of the well plate was covered with pre-prepared lead acetate-saturated filter paper and tightly sealed with a plastic lid. Subsequently, the assembly was incubated for 1 h at 37 °C in a 5% CO_2_ incubator. The filter paper was imaged using ChemiDoc^TM^ MP (Bio-Rad, Hercules, CA, USA) on a colorimetric setting. The spot intensities were quantified from images using GeneTools^®^, version 4.3.10.0 (Syngene, Cambridge, UK).

### 4.9. SDS-Page and Slot-Blot Immunoblotting

Samples of liver and kidney tissue were homogenized in ice-cold RIPA solution (Igepal ca-630, sodium deoxycholate, and 20% sodium dodecyl sulfate (SDS) in PBS) enriched with protease inhibitor cocktail (11836170001, Roche Diagnostics, Almere, The Netherlands), sodium orthovanadate (S6508, Sigma-Aldrich), and β-mercaptoethanol (805740, Sigma-Aldrich). Total protein concentration was determined using Bradford assay (5000116, Bio-Rad Laboratories, Veenendaal, The Netherlands) and 25 µg protein in 25 µL per sample was loaded onto a commercial gel (4568093, Bio-Rad). Proteins were separated by electrophoresis and transferred onto a nitrocellulose membrane using a Trans-Blot Turbo System (1,620,115 and 10,026,938 respectively, Bio-Rad Laboratories). Free nitrocellulose sites were blocked with 5% skim milk (70166, Sigma-Aldrich, Zwijndrecht, The Netherlands) for 20 min, and membranes were incubated overnight at 4 °C with anti-CBS antibody (1:1000 dilution, D8F2P, Cell Signaling Technology, Danvers, MA, USA). After three subsequent washes with TBST, membranes were incubated with secondary goat anti-rabbit antibody conjugated with HRP (1:2000, P0448, Dako, Glostrup, Denmark) at RT for 2 h. The chemiluminescent signal was detected using Western Lightning Ultra (NEL112001EA, Perkin Elmer International, Groningen, The Netherlands). The chemiluminescent signal was normalized to total lane protein volume using the standard BioRad StainFree TGX blot technology. For slot-blots, samples were diluted with TBS to a volume of 200 µL containing 2.5 µg protein. Proteins were transferred onto a nitrocellulose membrane using a 48-well Bio-Dot SF device (Bio-Rad Laboratories). Blots were blocked and washed as described for SDS-page blots and probed with anti-MDA antibody (anti-malondialdehyde antibody [11E3], ab243066, Abcam, Cambridge, UK). After three subsequent washes with TBST, membranes were incubated with secondary goat anti-mouse antibody conjugated with HRP (1:2000, P0447, Dako) at RT for 2 h, and imaged as described above.

### 4.10. Histology and Hematoxylin and Eosin Staining (HE)

During the termination procedure, after the animals were killed, organs were immediately placed in cold saline solution, portioned, and fixated in 4% formaldehyde solution for several days. After embedding in paraffin, 4 μm thick sections were cut (HistoCore AUTOCUT, Leica, Singapore) and placed on 76 × 26 mm glass slides (Starfrost, Brunswick, Germany).

Liver sections were deparaffinized and hydrated using an ethanol gradient (100%, 96%, and 70%, 10 min each). Sections were suspended in Mayer’s hematoxylin (5 min, MHS32, Sigma), washed with tap water, incubated in an eosin solution (5 min), and washed again. Sections were dehydrated and fixed in DePeX mounting resin (18243.01, Serva, VWR, Amsterdam, The Netherlands). HE-stained liver sections were scored for steatosis, lobular inflammation, and hepatocyte ballooning by a board-certified veterinary pathologist, based on the Kleiner scoring system as described previously [43].

### 4.11. Data Analysis and Statistics

Data were recorded in an electronic laboratory notebook eLABJOURNAL (Bio-ITech, Groningen, The Netherlands). Values in bar graphs are expressed as mean ± SD (standard deviation), *n* refers to the number of animals in each group. Unless otherwise specified, the differences between the groups were analyzed by either 2-way ANOVA (for parametric data) followed by Tukey’s multiple comparisons test, or Kruskal–Wallis testing (for non-parametric data) followed by Dunn’s method. Data analysis was performed using R version 4.0.2 and RStudio 1.3.1056. *p* values of less than 0.05 (two tailed) were considered statistically significant.

## Figures and Tables

**Figure 1 ijms-24-07019-f001:**
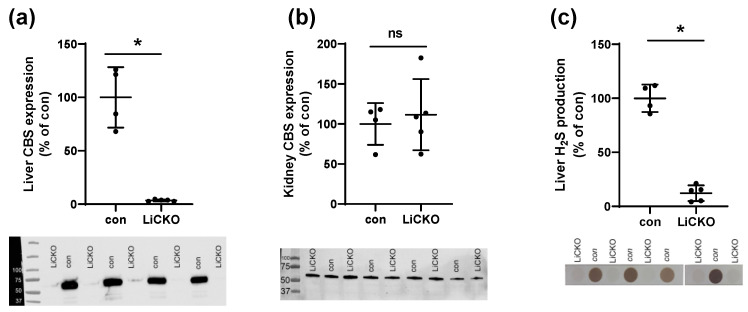
CBS is selectively deleted in liver. (**a**) CBS protein expression in LiCKO is significantly downregulated in liver; (**b**) CBS protein expression in LiCKO is unaffected in kidneys; (**c**) Down-regulation of CBS in liver is associated with impaired hydrogen sulfide production. * *p* < 0.05.

**Figure 2 ijms-24-07019-f002:**
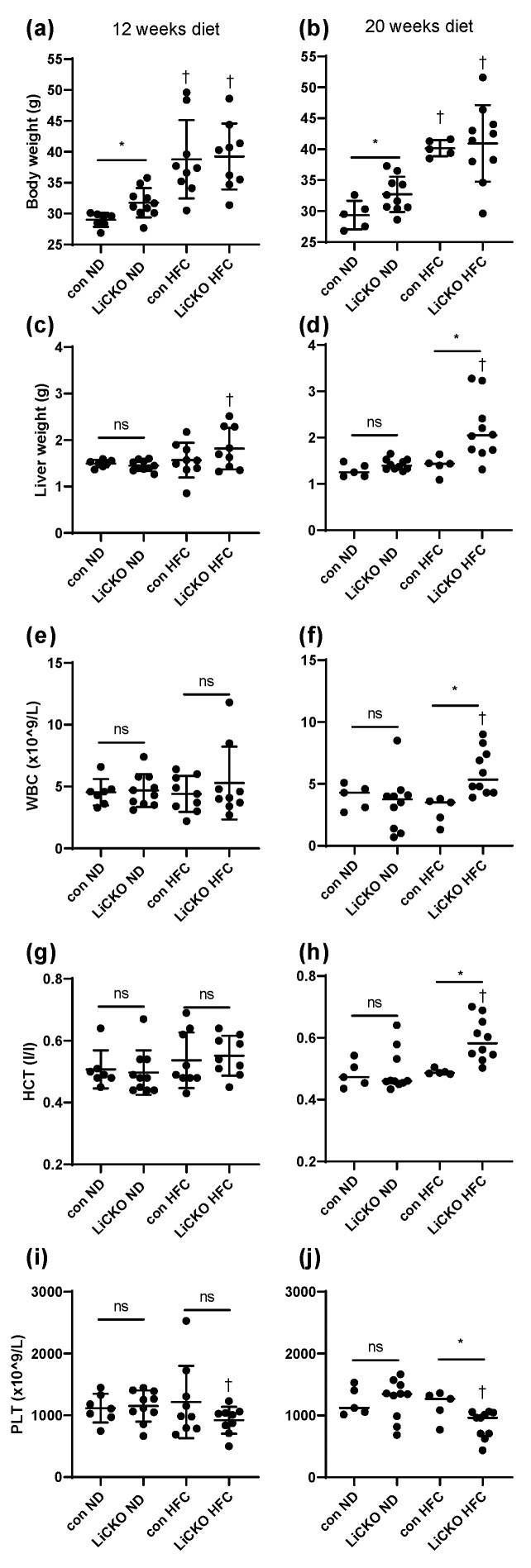
Body weight, liver weight, WBC, HCT, and PLT. (**a**,**b**) At 12 and 20 weeks of diet, body weight was higher in ND-fed LiCKO mice than controls on ND. Body weight of all groups on HFC diet were significantly increased compared with mice on ND, without differences between genotypes; (**c**) Liver weight of LiCKO mice on HFC diet was significantly increased over LiCKO on ND; (**d**) Liver weight of LiCKO mice on HFC diet was significantly increased over controls on HFC diet and over LiCKO on ND; (**e**) At 12 weeks of diet, no differences were observed in WBC; (**f**) At 20 weeks of diet, WBC in LiCKO mice on HFC diet was significantly increased over controls on HFC diet and over LiCKO on ND; (**g**) At 12 weeks of diet, no differences were observed in HCT; (**h**) At 20 weeks of diet, HCT in LiCKO mice on HFC diet was significantly increased over controls on HFC diet; (**i**) At 12 weeks of diet, no differences were observed in PLT; (**j**) At 20 weeks of diet, PLT in LiCKO mice on HFC diet was significantly decreased over controls on HFC diet and over LiCKO on ND; WBC; white blood cell count, HCT; hematocrit; PLT; platelet count; * *p* < 0.05 difference between con and LiCKO on same diet; † *p* < 0.05 difference between ND and HFC for con or LiCKO; ns = non-significant.

**Figure 3 ijms-24-07019-f003:**
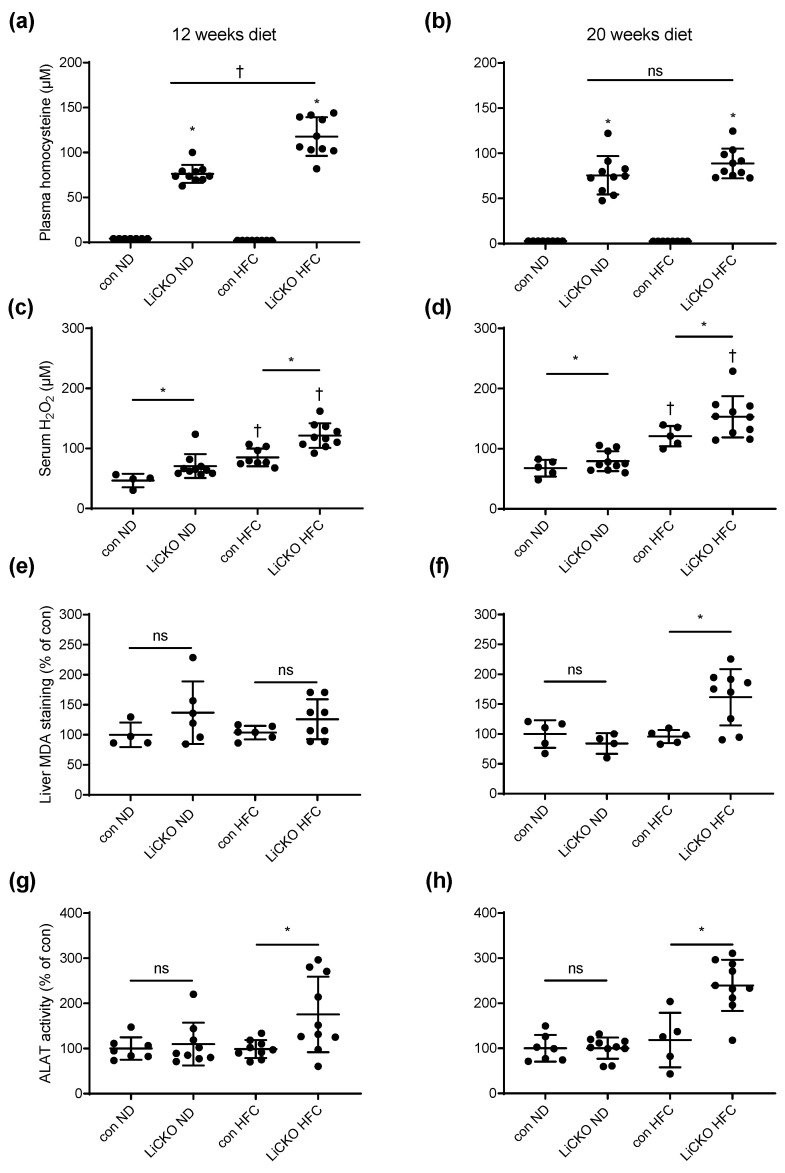
Plasma homocysteine, serum hydrogen peroxide, liver MDA staining, and plasma ALAT. (**a**) At 12 weeks of diet, homocysteine was elevated in plasma of ND-fed LiCKO mice and further significantly increased in LiCKO HFC mice; (**b**) At 20 weeks of diet, plasma homocysteine was similarly elevated in both LiCKO ND and LiCKO HFC; (**c**,**d**) At 12 and 20 weeks of diet, two-way ANOVA indicated that genotype (LiCKO) and diet (HFC) were both associated with further increased serum H_2_O_2_ concentrations; (**e**) No differences in liver MDA staining were observed between groups at 12 weeks of diet; (**f**) At 20 weeks of diet, liver MDA staining was significantly increased only in LiCKO HFC; (**g**,**h**) At 12 and 20 weeks, plasma ALAT activity was increased in LiCKO HFC over con HFC. * *p* < 0.05 difference between con and LiCKO on same diet; † *p* < 0.05 difference between ND and HFC for con or LiCKO; ns = non-significant.

**Figure 4 ijms-24-07019-f004:**
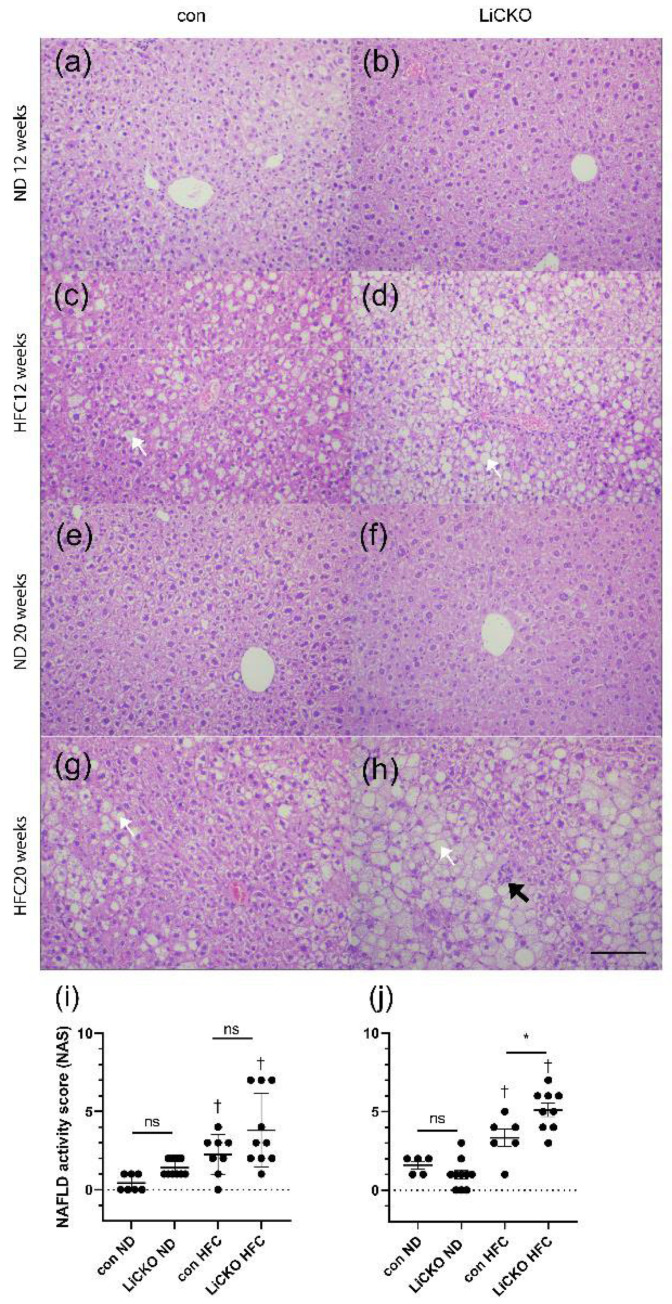
Liver histology. (**a**,**b**,**e**,**f**) No abnormal histological patterns were detected in mice on ND at 12 and 20 weeks of diet; (**c**) Control mice on HFC demonstrate mild steatosis (white arrow); (**d**) Steatosis is more aggravated in LiCKO at 12 weeks of HFC; (**g**) At 20 weeks of HFC, mild steatosis is present in control mice; (**h**) Further increased steatosis and inflammatory cells (black arrow) can be seen at 20 weeks of HFC in LiCKO; (**i**) NAFLD activity scores (NAS) at 12 weeks of diet; (**j**) NAFLD activity scores (NAS) were significantly increased in LiCKO mice at 20 weeks of HFC diet compared to controls on HFC diet. * *p* < 0.05 difference between con and LiCKO on same diet; † *p* < 0.05 difference between ND and HFC for con or LiCKO; ns = non-significant.

**Figure 5 ijms-24-07019-f005:**
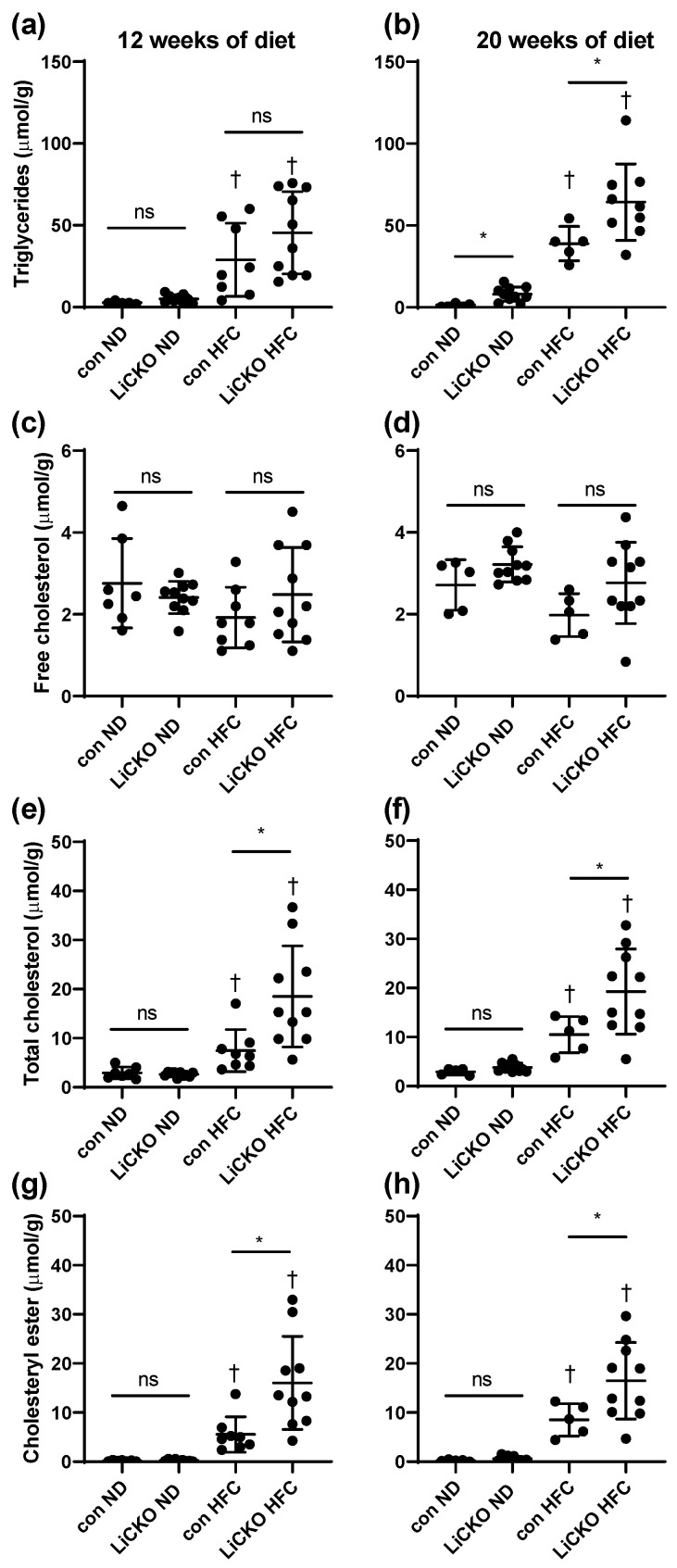
Liver triglyceride and cholesterol levels at weeks 12 and 20 of diet. (**a**,**b**) Liver triglyceride levels; (**c**,**d**) free cholesterol levels in liver; (**e**,**f**) total cholesterol and (**g**,**h**) cholesteryl levels. * *p* < 0.05 difference between con and LiCKO on same diet; † *p* < 0.05 difference between ND and HFC for con or LiCKO; ns = non-significant.

**Figure 6 ijms-24-07019-f006:**
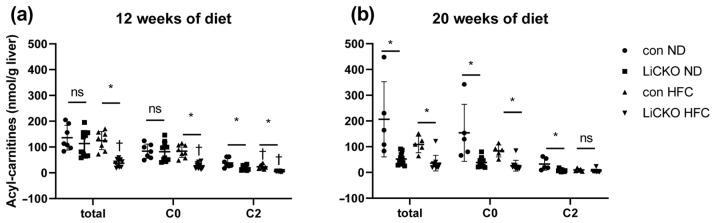
Acylcarnitines in liver. (**a**) Acylcarnitines at 12 weeks of diet; (**b**) Acylcarnitines at 20 weeks of diet; * *p* < 0.05 difference between con and LiCKO on same diet; † *p* < 0.05 difference between ND and HFC for con or LiCKO; ns = non-significant.

**Figure 7 ijms-24-07019-f007:**
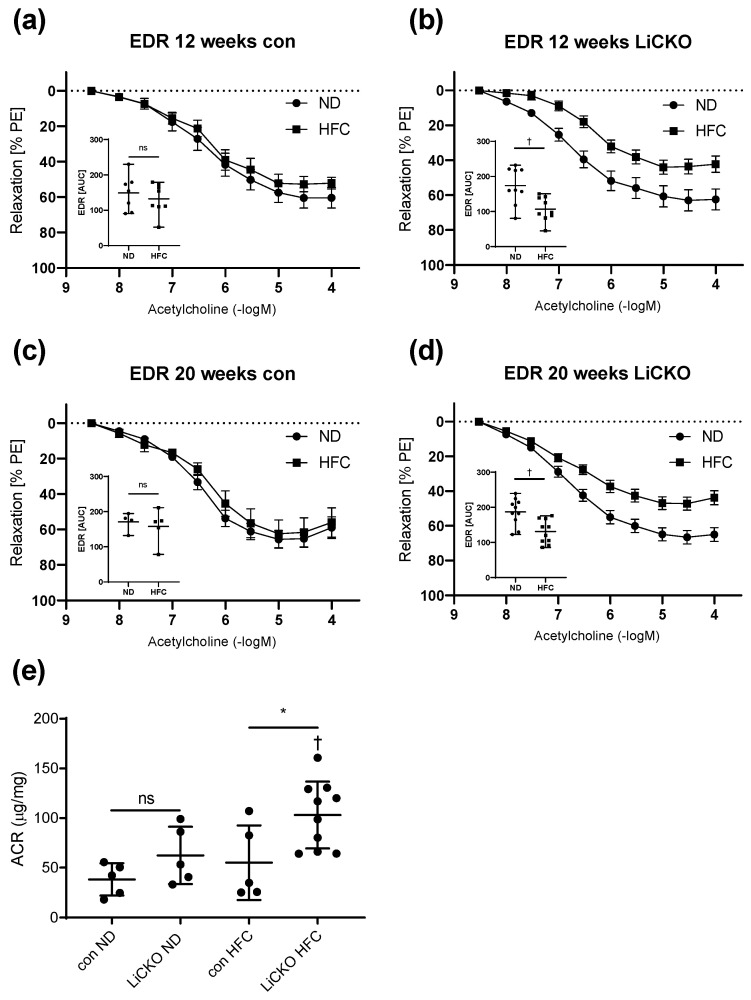
Effect of diet and genotype on vascular and endothelial function. (**a**,**c**) No effects of HFC diet on EDR were observed in control mice at 12 or 20 weeks of diet. Inserts show AUC for EDR; (**b**,**d**) Impaired EDR in LiCKO mice at both 12 and 20 weeks of HFC diet. Inserts show AUC for EDR; Quantification of the AUC demonstrated decreased EDR in LiCKO mice at both 12 and 20 weeks of HFC diet; (**e**) Increased ACR in urine of LiCKO mice at 20 weeks of HFC indicated renal endothelial damage. * *p* < 0.05 difference between con and LiCKO on same diet; † *p* < 0.05 difference between ND and HFC for con or LiCKO; ns = non-significant.

**Figure 8 ijms-24-07019-f008:**
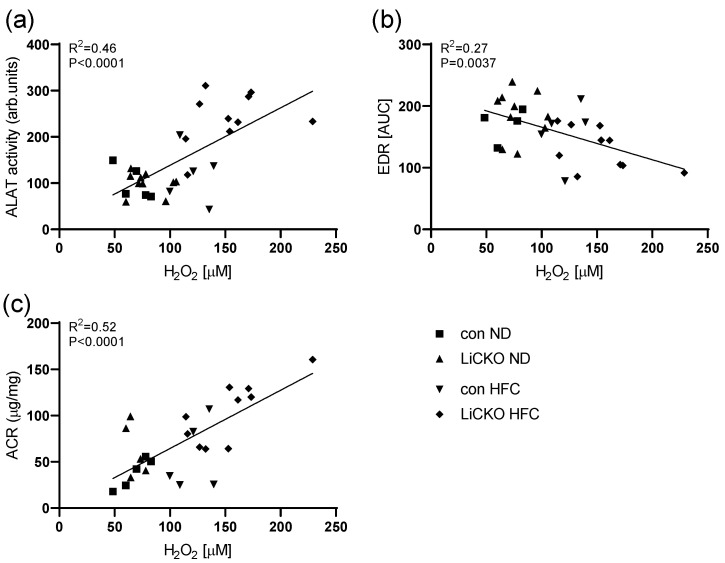
Correlations between liver and endothelial damage with oxidative state. (**a**,**c**) plasma ALAT and ACR activity correlated positively with serum H_2_O_2_ at week 20; (**b**) EDR correlated negatively with serum H_2_O_2_ at week 20.

**Table 1 ijms-24-07019-t001:** Animal characteristics.

	12 Weeks of Diet				20 Weeks of Diet			
Parameter	con ND	LiCKO ND	con HFC	LiCKO HFC	con ND	LiCKO ND	con HFC	LiCKO HFC
Body weight (g)	29 ± 0.4	**31.8 ± 0.8 ***	38.8 ± 2.1 †	39.2 ± 1.8 †	29.3 ± 1	**32.7 ± 0.9 ***	40.2 ± 0.6 †	40.9 ± 2 †
Heart (mg)	156.1 ± 3	**178.4 ± 6.5 ***	145.7 ± 8.6	147.2 ± 2.4 †	170.6 ± 14.4	164.4 ± 2.3	154.6 ± 7.5	173.6 ± 5.3
Left kidney (mg)	182.6 ± 5.4	191.9 ± 4.3	189 ± 6.8	181.9 ± 6.8	186.4 ± 14	197.5 ± 4.5	204.8 ± 15.9	191.1 ± 5.4
Right kidney (mg)	185.4 ± 5.7	201.6 ± 4.7	198.9 ± 5.6	193.7 ± 6.6	201 ± 13.6	209.9 ± 4.8	222 ± 6.8	214.2 ± 5.3
Spleen (mg)	75.7 ± 2.9	85.6 ± 4.5	83.8 ± 6.5	81.4 ± 4	77.6 ± 6.2	77.1 ± 2.4	94.2 ± 12.6	107.1 ± 17.9
Liver (mg)	1496 ± 29	1449 ± 38	1567 ± 125	1817 ± 149 †	1290 ± 63	1421 ± 38	1401 ± 89	**2167 ± 205 *†**
Brain (mg)	469.9 ± 4.2	483.2 ± 7.6	456.3 ± 9.7	463.4 ± 4.7	470.4 ± 7.7	469.4 ± 11.8	468.8 ± 12.3	469.5 ± 6.5
Glucose (mM)	13.4 ± 0.3	13.1 ± 0.4	13.6 ± 0.6	14 ± 0.4	13.5 ± 0.6	14.5 ± 0.7	13.6 ± 0.8	13.6 ± 0.7
WBC (×10^9^/L)	4.5 ± 0.4	4.7 ± 0.4	4.4 ± 0.5	5.3 ± 1	4 ± 0.5	3.5 ± 0.7	2.9 ± 0.5	**6 ± 0.6 *†**
RBC (×10^12^/L)	10.2 ± 0.5	9.8 ± 0.5	10.8 ± 0.6	11.1 ± 0.5	9.8 ± 0.4	9.7 ± 0.4	9.8 ± 0.1	**11.7 ± 0.4 *†**
HGB (mM)	9 ± 0.5	8.8 ± 0.4	9.4 ± 0.4	9.9 ± 0.4 †	8.7 ± 0.3	8.7 ± 0.4	8.8 ± 0.1	**10.2 ± 0.4 *†**
HCT (l/l)	0.51 ± 0.02	0.50 ± 0.02	0.54 ± 0.03	0.55 ± 0.02	0.48 ± 0.02	0.49 ± 0.02	0.49 ± 0.00	**0.59 ± 0.02 *†**
MCV (Fl)	49.7 ± 0.2	50.8 ± 0.5	49.5 ± 0.4	49.8 ± 0.3	49.2 ± 0.6	**51.1 ± 0.3 ***	50.1 ± 0.2	51 ± 0.3
MCH (amol)	883.3 ± 9.2	**896.9 ± 8.8 ***	872.4 ± 11.5	899.4 ± 10.8	893.2 ± 9.9	899.5 ± 4.5	895.8 ± 7.7	876.2 ± 6.8 †
MCHC (mM)	17.8 ± 0.2	17.7 ± 0.2	17.6 ± 0.3	18.1 ± 0.3	18.1 ± 0.2	**17.6 ± 0.1 ***	17.9 ± 0.2	**17.2 ± 0.1 *†**
PLT (×10^9^/L)	1116 ± 88	1151 ± 88	1216 ± 195	920 ± 72 †	1224 ± 102	1258 ± 102	1161 ± 108	**856 ± 70 *†**

WBC; white blood cell count, RBC; red blood cell count, HGB; hemoglobin, HCT; hematocrit, MCV; Mean corpuscular volume, MCH; mean corpuscular hemoglobin, MCHC; mean corpuscular hemoglobin concentration, PLT; platelet count. * *p* < 0.05 difference between con and LiCKO on same diet (also indicated in bold). † *p* < 0.05 difference between ND and HFC for con or LiCKO, ns = non-significant.

## Data Availability

Data are contained within this article or Appendix A.

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
