# Peer review of "Selective Hepatic Cbs Knockout Aggravates Liver Damage, Endothelial Dysfunction and ROS Stress in Mice Fed a Western Diet"

_ijms, 2023, doi:10.3390/ijms24087019_

Round 1
Reviewer 1 Report
This study investigates the importance of cystathionine -b-synthase in the liver for development of oxidative stress and hepatic inflammation and fatty liver disease. The model was developed by breeding homozygous Cbsflox/flox with hemizygous Alb-cre mice which resulted in a selective deficiency of CBS in the liver. Main findings included a worsening of signs of oxidative stress and damage and worsening of hepatic steatosis and inflammation. The study give additional important information about the central role of hepatic CBS function.
It is unclear what controls have been used. It is not stated if littermate controls or any C57BL/6 have been used as control. The exact nature of the control mice must be described.
The LiCKO mice showed increased WBC, RBC, Hb and HCT. However, platelet count was reduced. These findings are discussed in relation to systemic inflammation and findings in people with NASH and fibrosis. However, there may be other explanations such as reduced plasma volume to explained the increased WBC and RBC. Are there other signs of reduce plasma volume? The low platelet count could be explained by e.g. portal hypertension associated with splenomegaly. Considering that spleen weight was not significantly affected, splenomegaly affecting platelet count is not likely. Please expand the discussion about this topic.
In Scheme 2 body composition is described. In methods, it says that minispec LF was used. What is method is that? What is free fluid? Is it the same as extracellular water? Please use conventional terminology
Liver histology is not clearly described. The term NAS is often used to described the NASH activity, but it is not explained in the main manuscript. I suggest to give quantitative data on the liver histology in the main manuscript instead of referring to a pathology report. In the figure 3 legend, the black arrow indicates inflammation, but I assume it means inflammatory cells?
The description of unchanged fatty acid oxidation needs some clarification. Acylcarnitines were measured in the liver, while it is unclear if the isolated mitochondria were derived from the liver. The mitochondria respiration exps indicate no change in mitochondrial capacity but does not tell if the fatty acid oxidation is changed. As indicated reduced L-carnitine may result in reduced fatty acid oxidation capacity. However, reduced acetyl-carnitine has been associated with increased complete fatty acid oxidation. The Scheme 3 indicate, if anything, lower levels of longer chain acyl-carnitines in LiCKO mice indicating either more complete fatty acid oxidation or reduced flux/total fatty acid oxidation in the liver. However, RER as a measurement of whole body respiration was not affected in LiCKO mice as compared to controls. Please update the discussion on this topic, row 315 and onwards and discuss fatty acid oxidation separate from fatty acid oxidation capacity.
Line 291: Is it correct to use the word “ameliorated” in this sentence? Don’t think so.
Finally, it would have been interesting to investigate the effect of GYY4137 in this model.
Reviewer 2 Report
I have some minor comments to increase the quality of this publication.
In line 42 and 44, please add appropriate citations. in section 2.1, it would also be interesting to check for the H2O2 production levels and how they are related to the H2S levels.
The Protein ladder markers are missing in all the western blots and must be added. The authors should also include statistics for all the figures in the manuscript even if it is non-significant. it should be mentioned for consistency.
In section 2.2, the characteristics have been shown in the form of a table. However, it is very difficult to grasp the things from there in a short time for readers. the authors should make a bar chart for some the major characteristics so that it is easy yo check it and also include statistics.
There is no mention of figure 4a in the text.
On page 9, the figure is missing the legend.
In line 224, it is missing to mention that this is shown in figure 5b.
A correlation between H2O2 and H2S would be also very interesting to see and can be potentially added to figure 7 where the oxidative state is mentioned.
